# MicroRNAs and Metabolism: Revisiting the Warburg Effect with Emphasis on Epigenetic Background and Clinical Applications

**DOI:** 10.3390/biom11101531

**Published:** 2021-10-17

**Authors:** Zsuzsanna Gaál

**Affiliations:** Department of Pediatric Hematology-Oncology, Institute of Pediatrics, University of Debrecen, 4032 Debrecen, Hungary; gaal.zsuzsanna@med.unideb.hu

**Keywords:** cancer epigenetics, microRNAs, metabolism, Warburg effect, personalized treatment

## Abstract

Since the well-known hallmarks of cancer were described by Hanahan and Weinberg, fundamental advances of molecular genomic technologies resulted in the discovery of novel puzzle pieces in the multistep pathogenesis of cancer. MicroRNAs are involved in the altered epigenetic pattern and metabolic phenotype of malignantly transformed cells. They contribute to the initiation, progression and metastasis-formation of cancers, also interacting with oncogenes, tumor-suppressor genes and epigenetic modifiers. Metabolic reprogramming of cancer cells results from the dysregulation of a complex network, in which microRNAs are located at central hubs. MicroRNAs regulate the expression of several metabolic enzymes, including tumor-specific isoforms. Therefore, they have a direct impact on the levels of metabolites, also influencing epigenetic pattern due to the metabolite cofactors of chromatin modifiers. Targets of microRNAs include numerous epigenetic enzymes, such as sirtuins, which are key regulators of cellular metabolic homeostasis. A better understanding of reversible epigenetic and metabolic alterations opened up new horizons in the personalized treatment of cancer. MicroRNA expression levels can be utilized in differential diagnosis, prognosis stratification and prediction of chemoresistance. The therapeutic modulation of microRNA levels is an area of particular interest that provides a promising tool for restoring altered metabolism of cancer cells.

## 1. Introduction

Resulting from the advances in molecular biology, cell biology and genomics, the six well-known fundamental alterations of cancer, defined by Hanahan and Weinberg in 2000 (self-sufficiency in growth signals, insensitivity to growth inhibitory signals, evasion of programmed cell death, limitless replicative potential, sustained angiogenesis, tissue invasion and metastasis [1]), have been extended with novel hallmarks in 2011, which included genome instability, tumor-promoting inflammation, evading immune destruction and reprogramming energy metabolism [2]. However, Otto Warburg was awarded the Nobel Prize in 1931 for his discovery of cytochrome c oxidase, not for the formulation of the Warburg hypothesis [3], his observation that tumors take up and ferment high amounts of glucose to produce lactate even in the presence of oxygen [4,5] forms the basis of our current knowledge on the metabolic reprogramming of cancer cells.

Besides characterization of the unique metabolic phenotype of malignantly transformed cells, the better understanding of their epigenetic alterations is another cornerstone of current cancer research, casting new light on targeted therapeutic interventions. Pioneering work carried out by Miescher, Flemming, Kossel and Heitz between 1869 and 1928 resulted in the cytological distinction between regions of euchromatin and heterochromatin [6,7]. The first definition of epigenetics originated from Conrad Hal Waddington, who established this term in 1942 to describe inherited changes in phenotype without changes in the sequence of the DNA [8,9]. Four decades later, the first human disease to be linked to epigenetics was cancer. In 1983, Feinberg and Vogelstein described substantial hypomethylation in genes of cancer cells compared with their normal counterparts [10,11].

The recent explosion of our knowledge on epigenetic regulation highlights its importance in the pathogenesis of human cancer [12]. Epigenetic and metabolic alterations in cancer are not independent, but highly intertwined with each other [13]. Epigenetic modifiers require a series of metabolite cofactors, while the expression of metabolic enzymes is regulated by epigenetic mechanisms in a significant manner [14].

Since the first microRNA was identified in 1993 (transcribed from the Caenorhabditis elegans lin-4 locus [15]), it has become clear that microRNAs are strongly involved in the maintenance of both homeostatic chromatin structure and metabolic homeostasis. In this Review, we discuss the role of microRNAs in cancer epigenetics and their involvement in the altered metabolic phenotype of malignantly transformed cells, putting special emphasis on clinical applications and precision treatment approaches.

## 2. MicroRNAs and Carcinogenesis

MicroRNAs are a class of phylogenetically conserved, non-coding RNAs, with approximately 22–25 nucleotides in length [16,17]. During the multistep biogenesis of microRNAs, the long primary transcript is first trimmed into a hairpin-structured intermediate that is exported from the cell nucleus by the Exportin-5 transporter, in a Ran-GTP dependent manner [18]. In the cytoplasm, a miRNA:miRNA* duplex is formed, one strand of which is later incorporated into the effector complex named RNA-induced silencing complex (RISC) [19]. During the maturation of microRNAs, two consecutive cleavage steps are mediated by the RNase III endonuclease enzymes Drosha and Dicer [20]. MicroRNAs generally act as posttranscriptional repressors. They bind to the 3′ untranslated region (UTR) of the target mRNA [21], which is recognized by the microRNA seed sequence, located between the second and eighth nucleotides at their 5′ end [14]. Redundancy of the regulation is featured by the numerous targets of each microRNA, while a single mRNA can also be targeted by several different microRNAs [22].

The significant role of microRNAs during carcinogenesis is supported by increasing amounts of evidence [19]. The expression pattern of microRNAs distinguishes tumors of different developmental origin [23,24], therefore, their expression profiles can be utilized for the classification of human malignancies [25]. Alteration of microRNA expression levels in cancer was first reported in 2002 by Calin et al., when miR-15 and miR-16 were identified at 13q14.3, a frequently deleted region in chronic lymphocytic leukemia (CLL) [26]. During the next decade, it became clear that microRNAs are involved in the pathogenesis of all types of hematological malignancies and solid tumors [27]. This is in accordance with the finding that more than 50% of microRNA genes are located in cancer-associated genomic regions (CAGR) [28], and their targets include key regulators of cell cycle, proliferation, cell adhesion, apoptosis, angiogenesis and DNA repair [29,30].

MicroRNAs are involved in tumor initiation, progression and metastasis formation [31]. They are responsible for the regulation of interactions between cancer cells and cells of the tumor microenvironment including immune cells [15]. Furthermore, they can act either as tumor suppressors (anti-oncomiRs) or as oncogenic microRNAs (oncomiRs) by targeting oncogenes and tumor suppressor mRNAs, respectively [28]. However, certain microRNAs (e.g., miR-29) act as a tumor suppressor in leukemia and function as an oncomiR in solid tumors [31].

Impaired microRNA regulation of cell cycle progression contributes to the transformation of stem cells [32]. MicroRNAs regulate cyclin-dependent protein kinases (CDKs) and cyclines, moreover, the expression levels of numerous microRNAs vary between normal stem cells and cancer stem cells (CSCs) [32]. The miR-17-92 cluster cooperates with the c-Myc oncogene to prevent apoptosis in CSCs by targeting E2F and cyclin D [33], while the let-7 family of microRNAs has been shown to suppress epithelial–mesenchymal transition and other CSC characteristics by the regulation of numerous cell cycle components, such as CDK4, CDK6 and CDC25A [34]. MiR-377 and miR-145 are involved in the regulation of CSC properties in colon cancer and prostate cancer, respectively [35,36].

Disturbances of the microRNA biogenesis pathway have also been reported in numerous cancer types. Mutations in genes that encode Drosha, Dicer and Exportin-5, or mutations within the binding sites of target mRNAs, can contribute to the development of malignant diseases [15].

However, the expression of microRNAs is tightly controlled by transcription factors, microRNAs are also susceptible to epigenetic modulation [37]. Epigenetic inactivation by promoter hypermethylation has been detected in case of the tumor suppressor microRNAs miR-124 and miR-34a in hematological malignancies [38,39]. MicroRNAs regulate several enzymes of DNA methylation and histone modification, among which strong interconnections have been confirmed [40]. A typical example for the complexity of interactions is the regulatory loop between the anti-oncomiR miR-34a and the histone deacetylase enzyme SIRT1. MiR-34a mediates the repression of SIRT1, resulting in the inhibition of sterol-regulatory element-binding proteins (SREBPs) and nuclear factor κB (NFκB), while SIRT1 inhibits miR-34a by deacetylating its promoter [41]. The complex role of microRNAs during carcinogenesis is illustrated in Figure 1.

## 3. Implication of MicroRNAs in the Regulation of Metabolic Pathways

The first microRNA to be linked to metabolic regulation was miR-122 [42], which is expressed primarily in the liver, regulating lipid metabolism and liver cell differentiation [41]. Since then, growing number of microRNAs have been confirmed to regulate metabolic pathways [41]. MiR-33, contained by the primary transcript of SREBP2 (as an intronic microRNA), is involved in the regulation of cellular cholesterol export and fatty acid β-oxidation [43]. Furthermore, the α1 subunit of the nutrient and energy sensor AMP-dependent protein kinase (AMPK) is also targeted by this microRNA [41].

However, the best-characterized microRNAs in metabolic control are responsible for the maintenance of cholesterol and lipid homeostasis, and new data revealed the involvement of numerous microRNAs in the regulation of glucose homeostasis and insulin signaling as well [41]. MiR-103 and miR-107 regulate insulin and glucose homeostasis, while miR-223 controls the uptake of glucose in skeletal muscle by targeting glucose transporter 4 (GLUT4) [41]. MiR-375 was confirmed to be one of the key regulators of insulin secretion [41]. X component and B subunit of pyruvate dehydrogenase enzyme (PDH) are targeted by miR-26a [44] and miR-146b [45], respectively. Mitochondria has a central bioenergetic role due to encompassing important pathways of the carbohydrate, lipid and amino acid metabolism that are also under the control of microRNAs. The electron transport chain (ETC), tricarboxylic acid (TCA) cycle, fatty acid β-oxidation and amino acid metabolism are regulated by a large number of microRNAs, such as miR-210, miR-181a, miR-370 and miR-23a, respectively [46].

According to recently published data, the expression levels of metabolism-regulating microRNAs are modulated by a wide variety of environmental factors, including physical activity and nutrition. On a mouse model, altered microRNA expression profiles were detected in case of caloric restriction and high-fat diet [47]. Involvement of microRNAs in exercise adaptation has also been confirmed [48]. Maintenance of health requires the appropriate control of metabolic homeostasis [41]. A growing amount of evidence highlights the dysregulation of microRNAs in insulin resistance, diabetes, non-alcoholic fatty liver disease (NAFLD) and other metabolic disorders, in which specific microRNA signatures have been identified [41,49,50]. Based on the reversible and targetable changes of microRNA expression levels, their role in the metabolic reprogramming of cancer cells is an area of particular interest.

## 4. The Warburg Effect

In the late 1920s, Otto Warburg hypothesized that the glycolytic switch of cells causes cancer: “The origin of cancer lies in the anaerobic metabolic component of normal growing cells, which is more resistant to damage than is the respiratory component. Damage to the organism favours this anaerobic component and, therefore, engenders cancer” [51,52]. However, Warburg later proposed that mitochondrial dysfunctional is the root of aerobic glycolysis [53,54], and mitochondrial dysfunction promotes the Warburg effect only in a minority of tumors [55]. Metabolic alterations in proliferating cancer cells are induced by interactions between oncogenes and tumor suppressor genes that are also under the control of signaling cascades and microRNAs [56,57,58]. Therefore, glycolytic switch is considered to be an early event in oncogenesis [55] that is an outcome of oncogenic mutations [59].

The most well-known hallmark of the metabolic reprogramming of cancer cells is that the rate of glycolysis and lactate production is greatly increased, even in the presence of oxygen (aerobic glycolysis) [60,61]. The expression of the vast majority of glycolytic genes is regulated by c-Myc and hypoxia inducible factor 1α (HIF1α) transcription factors [62]. According to recently published data, aerobic glycolysis in Burkitt lymphoma cells is regulated by c-Myc, whereas in lymphoblastoid cell lines, HIF1α is responsible for the same phenomenon [63]. The hypoxia responsive elements (HRE) also encode for CXCR4 and CXCL12, which play an important role in the homing and preservation of leukemia stem cells [64]. High glycolytic rate is supported by some tumor-specific enzyme isoforms, such as hexokinase 2 (HK2) and pyruvate kinase M2 (PKM2) [65]. Besides the backup of glycolytic phospho-intermediates to be shuttled into biosynthetic pathways [55], non-metabolic functions of PKM2 have also been revealed that are essential for cell cycle progression and carcinogenesis [66]. Enhanced glycolysis is counterbalanced by the tumor suppressor p53 protein, that regulates the expression of TP53-induced glycolysis and apoptosis regulator (TIGAR), an enzyme responsible for decreasing the level of fructose-2,6-bisphosphate in cells [67].

Similarly to glycolytic enzymes, GLUT1, a rate-limiting factor for glucose uptake, is also aberrantly expressed in several tumor types [68]. Excessive glycolysis results in the excessive formation of lactate, which contributes to resistance against conventional therapies [55]. Tumor-derived lactate promotes the M2 polarization of tumor-associated macrophages, thereby suppressing anticancer immune response [61]. Lactate is excreted from cells by the lactate-proton symporter enzyme monocarboxylate transporter 4 (MCT4), expression of which is also under the control of HIF1α [69].

Sustained bioenergetic demand, required by uncontrolled proliferation, addicts cancer cells to an adequate anabolic supply [3]. The increased glucose consumption is used as a carbon source for de novo generation of nucleotides, proteins and lipids that can be diverted into multiple branching pathways, such as de novo biosynthesis of serine [61]. Substantially increased glutaminolysis serves as a major nitrogen source for proliferating cells, which also provides citrate to be utilized in fatty acid and cholesterol synthesis [70]. Upregulated activity of the pentose phosphate pathway (PPP) is essential for the high rate of nucleic acid synthesis, while the generation of NADPH provides a scavenger of reactive oxygen species (ROS) [71]. The production of ROS is elevated in malignantly transformed cells, resulting from increased metabolic rate and modifications in signaling pathways that affect cellular metabolism [66,72]. To counterbalance the higher production of ROS, the rate of ROS scavenging is also elevated in many tumors [66,73]. In contrast, the mutation of isocitrate dehydrogenase 1 (IDH1) enzyme results in the accumulation of lipid ROS, due to the reduced level of glutathione peroxidase 4 (GPX4) protein [74].

Besides losing their normal catalytic activity, mutant IDH1 and IDH2 gain the function of catalyzing the reduction of alpha-ketoglutarate (α-KG) to the oncometabolite 2-hydroxyglutarate (2-HG), that is a competitive inhibitor of numerous α-KG-dependent enzymes [75]. α-KG-dependent hydroxylases are a class of non-heme iron proteins, including TET enzymes of DNA hydroxymethylation, Jumonji-domain- (JMJD) containing histone demethylases and prolyl hydoxylase (PHD) enzymes that hydroxylate proline residues within the oxygen-dependent domains, resulting in the proteasomal degradation of the HIF1α transcription factor [76]. The PHD enzymes also have HIF1α-independent functions and among other factors, are also subject to regulation by the abnormal levels of oncometabolites that have been observed in many types of cancer [62]. According to recently published results, 2-HG accumulates in the extracellular space and is taken up by T lymphocytes, thereby compromising anticancer immune responses [77]. In case of cytogenetically normal acute myeloid leukemia (AML), high level of 2-HG was identified as a strong negative prognostic factor, independent of other molecular features [78]. Besides IDH, mutations of other TCA cycle enzymes, such as succinate dehydrogenase (SDH) and fumarate hydratase (FH), also lead to metabolic shifts of the cell due to the activation of HIF1α-mediated glucose utilization [57,79].

In summary, metabolic features of cancer cells can be distinguished as convergent and divergent metabolic phenotypes [80]. Enhanced glycolysis is the best example to convergent properties, which are shared among diverse types of tumors, while the stimulation of heterogeneous pathways results in divergent properties, such as the accumulation of 2-HG in case of IDH1 and IDH2 mutations [80]. Special metabolic symbiosis between cancer cells and cancer-associated stroma has also been described (referred to as the reverse Warburg effect), when glycolysis in the stromal cells supports adjacent cancer cells by the transfer of catabolites including lactate, pyruvate and ketone bodies [81].

## 5. MicroRNAs as Key Regulators of Cancer Metabolism—Epigenetic Background of the Warburg Effect

The strong intertwining between signaling molecules, oncogenes and tumor suppressor genes that are involved in the metabolic reprogramming of cancer cells [82] becomes an even more complex network by the regulation of these genes by microRNAs. The expression of c-Myc transcription factor is inhibited by p53-induced microRNAs miR-145 and miR-34c, while c-Myc was confirmed to activate the transcription of the oncogenic miR-17-92 cluster [58]. Based on its strong impact on glycolysis, TCA cycle and oxidative phosphorylation, the miR-17-92 cluster is considered to play a central role in the c-Myc driven metabolic reprogramming of cancer cells [83] (Table 1).

MiR-210, induced by the HIF1α, targets the mitochondrial iron–sulfur cluster assembly enzyme (ISCU) that provides cofactors for enzymes involved in the Krebs cycle and electron transport, therefore, the suppression of ISCU results in a shift to glycolysis under normoxic conditions [58,84].

A set of microRNAs (including miR-132, miR-144, miR-148b, miR-340 and miR-451) suppresses GLUT1-mediated glucose uptake, among which miR-132 was found to be downregulated in numerous types of cancer [85,86]. The tumor-specific HK2 isoform is regulated by both oncogenic and tumor suppressor microRNAs, such as miR-155 and miR-199a, respectively [82,87]. Another tumor-specific enzyme isoform, PKM2, is inhibited by miR-326 and miR-122, the latter of which is considered to be a tumor suppressor microRNA that decreases the occurrence of metastasis in hepatocellular carcinoma via the downregulation of PKM2 [88,89].

Lactate dehydrogenase (LDH) is a tetrameric enzyme, the subunits of which are encoded by two different genes, LDH-A and LDH-B. Elevated LDH-A/LDH-B ratio is characteristic to tumor cells and promotes lactate formation in a significant manner. While HIF1α- and c-Myc-related pathways promote the expression of LDH-A [90], overexpression of the tumor suppressor miR-34a counteracts this effect [91]. Similarly to miR-34a, miR-422 also inhibits the Warburg effect. Activity of pyruvate dehydrogenase (PDH) can be restored via the suppression of pyruvate dehydrogenase kinase 2 (PDK2) by miR-422, that was found to be downregulated in gastric cancer [92]. On the other hand, miR-26a promotes the Warburg effect by targeting the X component of PDH, and thereby inhibiting the key step of glycolysis entry into the TCA cycle [44].

Glutamine metabolism and enzymes of the PPP are also regulated by microRNAs. The alanine/serine/cysteine-preferring transporter 2 (ASCT2) of glutamine, upregulated in different kinds of cancer, is targeted by miR-137 [93], while the repression of miR-23a and miR-23b resulted in a higher expression level of glutaminase enzyme (GLS) [94]. ATP citrate lyase (ACLY), which is a key enzyme of de novo fatty acid synthesis, was found to be upregulated in numerous types of cancer, and it is inhibited by miR-22 in osteosarcoma and lung cancer cells [95]. Fatty acid synthase (FASN) is a central lipogenic enzyme that is targeted by miR-15 and miR-16. FASN was found to be upregulated in breast cancer [96]. MiR-1, miR-122 and miR-206 negatively regulate the expression of glucose-6-phosphate dehydrogenase (G6PD), the enzyme that catalyzes the first reaction of PPP [71]. Transketolase, involved in the non-oxidative phase of PPP, is targeted by miR-497, a microRNA that modulates cisplatin chemosensitivity of cervical cancer cells [97,98].

Besides microRNAs, DNA-methylation and a wide variety of histone modifications also contribute to the metabolic reprogramming of cancer cells, thereby composing a complex epigenetic background of the Warburg effect. Promoter methylation of the glycolysis antagonist fructose-1,6-bisphosphatase-1 (FBP1) is promoted by the NFκB pathway, and can be used as a biomarker for prognosis prediction in gastric cancer [99]. The promoter hypermethylation of LDH-B, detected in breast and prostate cancer, can be restored by the demethylating agent 5-azacytidine [100]. Methylation of PKM2 by coactivator-associated arginine methyltransferase 1 (CARM1) at three arginine residues results in the localization of PKM2 to the mitochondria-associated endoplasmic reticulum membrane, which promotes aerobic glycolysis by decreasing Ca^2+^-uptake and mitochondrial membrane potential [101]. Promoter hypermethylation of Derlin-3, which is implicated in GLUT1 proteasome degradation, contributes to the overexpression of GLUT1 transporter [102].

Monoubiquitination of histone H2B (H2Bub1) exerts an anti-Warburg effect by regulating the expression of mitochondrial respiratory genes. In addition, PKM2 interacts with H2B and decreases the level of H2Bub1 [103]. Members of the NAD^+^-dependent sirtuin family of histone deacetylase enzymes play an important role in the metabolic regulation of cancer cells [104,105]. In tumor cell lines, the absence of SIRT3 led to the overproduction of ROS, resulting in the stabilization of HIF1α and the upregulation of its glycolytic targets [106]. SIRT4 enzyme has been identified as a tumor suppressor and glutamine gatekeeper, which inhibits the glutamate dehydrogenase (GDH) enzyme [107]. In mouse embryonic fibroblasts, loss of SIRT4 enzyme resulted in increased glutamine-dependent proliferation and stress-induced genomic instability [108,109]. Recently published data also suggest the antitumor activity of the HIF1α corepressor SIRT6 enzyme. Besides its impact on the expression of glycolytic genes, SIRT6 also regulates the splicing of the tumor-specific PKM2 isoform [110].

All in all, besides the protein-coding oncogenes and tumor suppressor genes, epigenetic regulatory mechanisms such as DNA methylation, sirtuin enzymes and microRNAs are also key regulators of the Warburg effect, providing an unprecedented scale of potential therapeutic targets.

## 6. Clinical Applications

Advances in molecular biology and genomics opened up new horizons for anticancer treatment, targeting metabolic pathways, microRNAs and epigenetic regulators. Chemo- and radiotherapy-resistant breast cancer cells were re-sensitized by 2-deoxyglucose, a competitive inhibitor of glucose [111]. WZB117, an inhibitor of GLUT1 [112], was confirmed to exert synergistic effects with paclitaxel and cisplatin [65]. Pharmacological inhibition of GLUT1 with BAY-876 impaired the growth of triple-negative breast cancer cells [113], while the inhibition of GLUT3-transporter resulted in delayed resistance to temozolomide in the treatment of glioblastoma [114].

Co-administration of glucocorticoids with inhibitors of HK2, such as 3-bromopyruvate, increased in vitro sensitivity of glucocorticoids in acute lymphoblastic leukemia (ALL) [115]. In human lung cancer cells, the silencing of PKM2 resulted in the increased efficacy of docetaxel in vitro and in vivo [116]. CPI-613 (devimistat) is a lipoate analog which inhibits PDH and α-KG dehydrogenase complexes [66]. CPI-613 re-sensitized AML cells to cytotoxic agents through the inhibition of TCA cycle [66]. While no specific inhibitors of wild-type IDH enzyme have been reported, potent inhibitors have been identified for mutant IDH1 and mutant IDH2. AGI-5198 was reported to inhibit the accumulation of 2-HG in IDH1-mutated glioma cells in vivo [117], that also promoted the differentiation of glioma cells [118]. In preclinical studies, epigallocatechin gallate was confirmed to interrupt the anaplerotic use of glutamine in the TCA cycle, thereby reducing tumor growth [119].

BPTES (*bis*-2-(5-phenylacetamido-1,2,4-thiadiazol-2-yl)-ethyl-sulfide) is an allosteric inhibitor of GLS1 enzyme. In pancreatic carcinoma, the synergistic effect of BPTES and doxorubicin was observed [59]. While ALL cells have an increased dependence on exogenous asparagine due to the decreased activity of asparagine synthetase enzyme [120], AML cell lines that are resistant to cytarabine therapy show a significant alteration to purine metabolism [121]. Etomoxir (irreversible inhibitor of carnitine palmitoyltransferase-1) inhibited cell viability in glioblastoma cells with a significant reduction in ATP and NADPH levels [122]. Sensitivity against different kinds of antimetabolites also showed associations with cytogenetic properties. Comparing AML cell lines, NB4 (acute promyelocytic leukemia with t(15;17) translocation) and THP-1 (acute monocytic leukemia with t(9;11) translocation) cells exhibited increased sensitivity to 2-deoxyglucose and etomoxir, respectively [123].

Modulating the expression levels of microRNAs is receiving a great deal of attention in cancer. Two major fields are replenishing the expression of anti-oncomiRs, and targeting oncomiRs with microRNA antagonists (antimiRs) [124]. MicroRNA mimics are double-stranded molecules matching the corresponding microRNA sequence, while antimiRs have a single-stranded structure, classified as first-generation antisense oligonucleotides (ASOs) and locked nucleic acids (LNAs) [15]. There are several ongoing clinical trials such as the LNA-modified antimiR-155 in cutaneous T cell lymphoma and the miR-34 mimic lipid nanoparticles in multiple solid tumors [15]. Besides the growing number of therapeutic targets, the spectrum of delivery systems for microRNA therapeutics has also broadened and includes neutral lipid emulsions, dendrimers (polyamidoamine- or polypropylene imine-conjugated nucleic acids), cyclodextrin (glucose polymer), adenoviral vectors, polylactide-co-glycolide (PLGA) polymers, chitosan (cationic polymer derived from chitin) and bacterium-derived EnGeneIC Delivery Vehicle (EDV) nanocells (also called targomiRs) [15]. Targeting a wide variety of microRNAs (such as miR-122, miR-125b, miR-34a, miR-155 and miR-205) with these novel technologies can potentially normalize dysregulated metabolic enzymes in chemoresistant cancer cells [125]. Numerous microRNAs have been confirmed to enhance the efficacy of anticancer drugs [124]. Overexpression of the GLUT1-targeting miR-218 increased chemosensitivity of bladder cancer cells to cisplatin [86], while miR-153 enhanced sensitivity against arsenious acid in chronic myeloid leukemia (CML) [126]. Dietary microRNAs represent a new area and are released into the circulation after cellular uptake in the gastrointestinal tract. They are transported to multiple cell types and tissues, such as liver and immune cells, to directly regulate gene expression [127].

There are further promising clinical applications of microRNAs, including differential diagnosis, prediction of prognosis and chemoresistance. Based on the expression level of four microRNAs (miR-128a, miR-128b, let-7b and miR-223), ALL and AML can be distinguished with high accuracy (97–99%) [128]. MicroRNA expression levels can be utilized in the early detection of bladder cancer [129] and in the differential diagnosis of non-small cell lung carcinoma [130]. Let-7a and miR-188 are prognostic biomarkers in cytogenetically normal AML [131]. A six-microRNA-based model has been described to improve prognosis prediction in breast cancer [132], and a five-microRNA-based signature was identified to have a significant prognostic value in colon cancer [133]. Downregulation of miR-181a was associated with cytarabine resistance in HL60 cells, due to reduced targeting of the BCL2 oncogene [134]. In glioma cell lines, miR-16 was confirmed to modulate temozolomide resistance by regulating BCL2 [135]. MicroRNAs have also been identified in exosomes, which can be taken up by neighboring or distant cells [136]. Exosomal microRNAs are involved in cancer progression and metastasis formation [136], they are ligands of toll-like receptors and activate immune cells [137]. A special circular RNA, hsa_circ_0005963 (ciRS-122), is a sponge for the PKM2-targeting miR-122. Exosomes from oxaliplatin-resistant colorectal cancer cells delivered ciRS-122 to sensitive cells, thereby promoting glycolysis and chemoresistance [138]. Clinical applications of microRNAs and therapeutic approaches to counteract Warburg effect are summarized in Figure 2.

## 7. Concluding Remarks and Future Perspectives

Epigenetic and metabolic alterations have been characterized as novel hallmarks of cancer during the past two decades. Better understanding of these features became a top priority of cancer research, which also revealed their multiple interactions with previously already well-characterized etiologic factors such as disturbances of signaling cascades, mutations in oncogenes and tumor suppressor genes. Mutations of epigenetic modifiers have been identified as early events in several cancers. The epimutation concept proposes the vertical transmission of an error-prone epigenetic pattern, resulting in the generation of clones with abnormal mitoses and malignant characteristics [139]. Altered epigenetic profile and metabolic reprogramming of cancer cells provide promising novel therapeutic targets. However, special challenges of such novel therapeutic approaches should be noted.

Treatments targeting altered metabolic phenotypes should consider the heterogeneity of metabolism between different types of tumors and even within distinct regions of a solid tumor [80]. Divergent metabolic properties (such as the accumulation of 2-HG) can be targeted with a more acceptable toxicity profile compared to that of convergent metabolic phenotypes [80]. Targeting CSC metabolism is also a promising treatment option in cancer. An increasing amount of evidence suggests the metabolic plasticity of CSCs, which contributes to their resistance to conventional therapies [140]. CSCs can favor glycolysis or oxidative phosphorylation, depending on the niche where they are located [141]. Based on the rapid transition of the metabolic phenotype of CSCs under glucose deprivation or hypoxia, targeting the adaptive mechanisms is an optional treatment approach [141].

Ensuring appropriate specificity is a topmost challenge of epigenetic therapies. Epigenetic drugs should be transmitted specifically to distinct regions of chromatin. Growing number of delivery vehicles and a wide variety of modified oligonucleotides have recently been constructed that can be utilized in the modulation of microRNA expression levels. MicroRNAs affect hundreds of targets in complex regulatory networks. Therefore, the context-dependent role of microRNAs should always be taken into consideration. Besides the therapeutic modulation of microRNA expression levels, a wide variety of their further clinical applications should be highlighted, including differential diagnosis, biomarkers for advanced prognostic stratification and prediction of chemoresistance. While the modulation of microRNA levels can enhance chemosensitivity, the impact of conventional chemotherapeutic agents on microRNAs also should be noted. For example, 5-fluorouracil has been shown to enhance the expression of anti-oncomiRs such as the let-7 family and miR-15b [134]. In recent years, the number of clinical trials targeting Warburg-related microRNAs has increased. Though the majority of these drugs were well tolerated, the possibility of unpredictable side effects is highlighted by the fact that the trial of MRX34 (miR-34 mimic molecule targeting LDH-A) was terminated due to severe immune-related reactions [142].

Early identification of reversible epigenetic and metabolic alterations is of key significance to increase the efficacy and reduce the toxicity of cancer treatment by advanced prognosis stratification and novel combinations of conventional and targeted therapeutic agents.

## Figures and Tables

**Figure 1 biomolecules-11-01531-f001:**
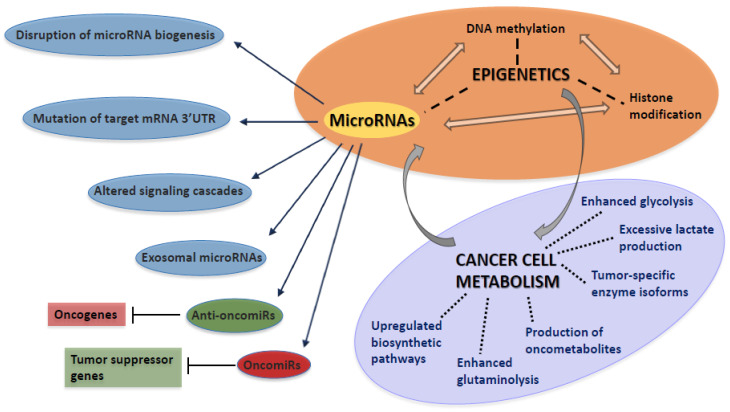
MicroRNAs contribute to the initiation and progression of cancer by a wide variety of different mechanisms, including multiple interactions with oncogenes, tumor suppressor genes, DNA methylation and histone modification. Altered microRNA expression levels are also involved in the metabolic reprogramming of cancer cells. Targeting such Warburg-related microRNAs is a promising therapeutic approach. UTR: untranslated region.

**Figure 2 biomolecules-11-01531-f002:**
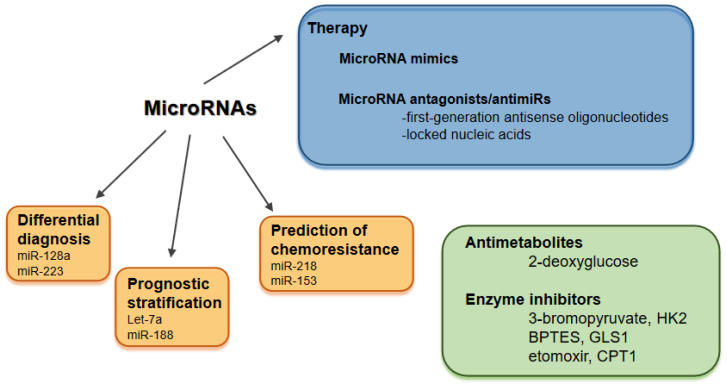
Clinical applications of microRNAs and therapeutic approaches to counteract Warburg effect. Abbreviations: BPTES: *bis*-2-(5-phenylacetamido-1,2,4-thiadiazol-2-yl)-ethyl-sulfide, CPT1: carnitine palmitoyltransferase-1, GLS: glutaminase enzyme, HK2: hexokinase 2 isoform.

**Table 1 biomolecules-11-01531-t001:** MicroRNAs are involved in both the maintenance of metabolic homeostasis and the metabolic reprogramming of cancer cells. Examples for Warburg-promoting and anti-Warburg microRNAs are highlighted with blue and green background, respectively.

MicroRNA	Implication in Metabolic Regulation and Targets
miR-1	G6PD
miR-15	BCL2, FASN
miR-16	BCL2, FASN
miR-17-92 cluster	glycolysis, TCA, oxidative phosphorylation, E2F, cyclin D
miR-22	ACLY
miR-23a	amino acid metabolism, GLS
miR-23b	amino acid metabolism, GLS
miR-26a	pyruvate–lactate conversion, PDH X component
miR-33	fatty acid β-oxidation
miR-34a	LDH-A, SIRT1
miR-122	lipid metabolism, PKM2
miR-103	insulin and glucose homeostasis
miR-107	insulin and glucose homeostasis
miR-132	GLUT1
miR-137	ASCT2
miR-144	GLUT1
miR-146b	pyruvate–lactate conversion
miR-181a	TCA
miR-155	HK2
miR-199a	HK2
miR-206	G6PD
miR-210	electron transport chain, glycolytic enzymes, ISCU
miR-223	GLUT4
miR-326	PKM2
miR-370	fatty acid β-oxidation
miR-375	insulin secretion
miR-422	PDK2
miR-451	GLUT1
miR-497	transketolase

Abbreviations: ACLY: ATP citrate lyase; ASCT2: alanine/serine/cysteine-preferring transporter 2; BCL2: B-cell lymphoma 2 gene; CDK: cyclin-dependent protein kinase; FASN: fatty acid synthase; G6PD: glucose-6-phosphate dehydrogenase; GLS: glutaminase; GLUT: glucose transporter; HK2: hexokinase 2 isoform; ISCU: iron–sulfur cluster assembly enzyme; LDH: lactate dehydrogenase; PDH: pyruvate dehydrogenase; PDK2: pyruvate dehydrogenase kinase 2; PKM2: pyruvate kinase M2 isoform; SIRT: sirtuin enzyme; TCA: tricarboxylic acid cycle.

## Data Availability

No new data were created or analyzed in this study. Data sharing is not applicable to this article.

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
