# Peer review of "MicroRNAs and Metabolism: Revisiting the Warburg Effect with Emphasis on Epigenetic Background and Clinical Applications"

_biomolecules, 2021, doi:10.3390/biom11101531_

Round 1

Reviewer 1 Report

  1. The authors have explained some of the good concepts in microRNAs vs metabolism vs  Warburg Effect. The review is interesting and adds more information to this research area. 
  2. I would recommend listing out the number of miRNAs that fall in this area and provide them in the form of a Table
  3. The majority of the articles the authors cited in this paper are very old and I would recommend them to cite the latest ones
  4. The conclusion drawn from the manuscript needs to be elaborated. 

Author Response

  The authors have explained some of the good concepts in microRNAs vs metabolism vs  Warburg Effect. The review is interesting and adds more information to this research area. 

  I would recommend listing out the number of miRNAs that fall in this area and provide them in the form of a Table

Table 1 has been extended with further microRNAs that fall in this area and table legend was completed with all of the abbreviations that are indicated in the table (page 8 of the revised manuscript).

MicroRNA

Implication in metabolic regulation and targets

miR-1

G6PD

miR-15

BCL2, FASN

miR-16

BCL2, FASN

miR-17-92 cluster

glycolysis, TCA, oxidative phosphorylation, E2F, cyclin D

miR-22

ACLY

miR-23a

amino acid metabolism, GLS

miR-23b

amino acid metabolism, GLS

miR-26a

pyruvate-lactate conversion, PDH X component

miR-33

fatty acid β-oxidation 

miR-34a

LDH-A, SIRT1

miR-122

lipid metabolism, PKM2

miR-103

insulin and glucose homeostasis

miR-107

insulin and glucose homeostasis

miR-132

GLUT1

miR-137

ASCT2

miR-144

GLUT1

miR-146b

pyruvate-lactate conversion

miR-181a

TCA

miR-155

HK2

miR-199a

HK2

miR-206

G6PD

miR-210

electron transport chain, glycolytic enzymes, ISCU

miR-223

GLUT4

miR-326

PKM2

miR-370

fatty acid β-oxidation 

miR-375

insulin secretion

miR-422

PDK2

miR-451

GLUT1

miR-497

transketolase

Table 1. MicroRNAs are involved in both the maintenance of metabolic homeostasis and the metabolic reprogramming of cancer cells. Examples for Warburg-promoting and anti-Warburg microRNAs are highlighted with blue and green background, respectively.

Abbreviations: ACLY: ATP citrate lyase, ASCT2: alanine/serine/cysteine-preferring transporter 2, BCL2: B-cell lymphoma 2 gene, CDK: cyclin-dependent protein kinase, FASN: fatty acid synthase, G6PD: glucose-6-phosphate dehydrogenase, GLS: glutaminase, GLUT: glucose transporter, HK2: hexokinase 2 isoform, ISCU: iron-sulfur cluster assembly enzyme, LDH: lactate dehydrogenase, PDH: pyruvate dehydrogenase, PDK2: pyruvate dehydrogenase kinase 2, PKM2: pyruvate kinase M2 isoform, SIRT: sirtuin enzyme, TCA: tricarboxylic acid cycle

  • The majority of the articles the authors cited in this paper are very old and I would recommend them to cite the latest ones

To address the request to cite the latest papers, 21 novel references were added in the revised version, that are indicated with yellow in the reference list. The majority of these papers were published between 2018 and 2020. Please note, that the additional references resulted in a change in the numbering of citations.

  The conclusion drawn from the manuscript needs to be elaborated. 

In the section of conclusions, some additional issues have been discussed in connection with mutations of epigenetic modifiers, metabolic plasticity of cancer stem stells, clinical trials targeting Warburg-related microRNAs, and the interplay between conventional chemotherapeutic agents and microRNAs:

„Mutations of epigenetic modifiers have been identified as early events in several cancers. The epimutation concept proposes the vertical transmission of an error-prone epigenetic pattern, resulting in the generation of clones with abnormal mitoses and malignant characteristics [142]” – lines 388-391 of the revised manuscript.

„Targeting CSC metabolism is also a promising treatment option in cancer. Increasing amount of evidence suggests the metabolic plasticity of CSCs, that contributes to their resistance to conventional therapies [143]. CSCs can favor glycolysis or oxidative phophorylation, depending on the niche where they are located [144]. Based on the rapid transition of the metabolic phenotype of CSCs under glucose deprivation or hypoxia, targeting the adaptive mechanisms is an optional treatment approach [144]” – lines 397-402 of the revised manuscript.

„While the modulation of microRNA levels can enhance chemosensitivity, the impact of conventional chemotherapeutic agents on microRNAs also should be noted. For example, 5-fluorouracil has been shown to enhance the expression of anti-oncomiRs such as the let-7 family and miR-15b [137]. In the recent years, number of clinical trials targeting Warburg-related microRNAs has increased. Though the majority of these drugs were well tolerated, the possibility of unpredictable side effects is highlighted by the fact that the trial of MRX34 (miR-34 mimic molecule targeting LDH-A) was terminated due to severe immune-related reactions [145]” - lines 410-417 of the revised manuscript.

Reviewer 2 Report

In the review “MicroRNAs and Metabolism: Revisiting the Warburg Effect with Emphasis on Epigenetic Background and Clinical Applications” authors discuss the role of microRNAs in the altered epigenetic pattern and metabolic phenotype of cancer. They also describe microRNAs connections with oncogenes, tumor suppressor genes and epigenetic modifiers. The metabolic reprogramming of cancer cells results from the dysregulation of a complex network, in which the microRNAs are found in the central hubs. MicroRNAs regulate the expression of different metabolic enzymes including tumor specific isoforms, they also influence the epigenetic pattern. Authors also  describe a role of MicroRNA expression in differential diagnosis, prognosis stratification, and predicting chemoresistance.

The paper is well developed, images have proper descriptions and references are suitable.

In my opinion the manuscript could be accepted for publication after introducing the following minor changes:

-Lines 197-199: “In contrast, as a consequence of the mutation of isocitrate dehydrogenase 1 (IDH1) enzyme, reduced level of glutathione peroxidase 4 (GPX4) protein was observed, that is a key enzyme of lipid ROS removal [63].”

The sentence is not clear, please rearrange the sentence to better conclude the paragraph.

-Authors extensively describe the role of miRNA and its involvement in cancer, however the issue concerning the  transformation caused by miRNA in stem cells and their dysregulation toward a malignant phenotype is not mentioned. Please develop this point concerning stem cells behavioural changing involving miRNA.

Author Response

In the review “MicroRNAs and Metabolism: Revisiting the Warburg Effect with Emphasis on Epigenetic Background and Clinical Applications” authors discuss the role of microRNAs in the altered epigenetic pattern and metabolic phenotype of cancer. They also describe microRNAs connections with oncogenes, tumor suppressor genes and epigenetic modifiers. The metabolic reprogramming of cancer cells results from the dysregulation of a complex network, in which the microRNAs are found in the central hubs. MicroRNAs regulate the expression of different metabolic enzymes including tumor specific isoforms, they also influence the epigenetic pattern. Authors also  describe a role of MicroRNA expression in differential diagnosis, prognosis stratification, and predicting chemoresistance.

The paper is well developed, images have proper descriptions and references are suitable.

In my opinion the manuscript could be accepted for publication after introducing the following minor changes:

-Lines 197-199: “In contrast, as a consequence of the mutation of isocitrate dehydrogenase 1 (IDH1) enzyme, reduced level of glutathione peroxidase 4 (GPX4) protein was observed, that is a key enzyme of lipid ROS removal [63].”

The sentence is not clear, please rearrange the sentence to better conclude the paragraph.

Yes, I agree with the reviewer that this sentence was not clear, therefore, it has been modified as the following (lines 211-213 of the revised manuscript):

„In contrast, the mutation of isocitrate dehydrogenase 1 (IDH1) enzyme results in the accumulation of lipid ROS, due to the reduced level of glutathione peroxidase 4 (GPX4) protein [76]”.

Please note, that some additional references were inserted in the revised manuscript, that resulted in a change in the numbering of citations.

-Authors extensively describe the role of miRNA and its involvement in cancer, however the issue concerning the transformation caused by miRNA in stem cells and their dysregulation toward a malignant phenotype is not mentioned. Please develop this point concerning stem cells behavioural changing involving miRNA.

In order to highlight the importance of micoRNAs in the altered behaviour and malignant transformation of stem cells, a new paragraph was added to the section of „MicroRNAs and carcinogenesis” – lines 110-117 of the revised manuscript:

„Impaired microRNA regulation of cell cycle progression contributes to the transformation of stem cells [32]. MicroRNAs regulate cyclin-dependent protein kinases (CDKs) and cyclines, moreover, the expression levels of numerous microRNAs vary between normal stem cells and cancer stem cells (CSCs) [32]. The miR 17-92 cluster cooperates with the c-myc oncogene to prevent apoptosis in CSCs by targeting E2F and cyclin D [33], while the let-7 family of microRNAs has been shown to suppress epithelial-mesenchymal transition and other CSC characteristics by the regulation of numerous cell cycle components, such as CDK4, CDK6 and CDC25A [34]. MiR-377 and miR-145 are involved in the regulation of CSC properties in colon cancer and prostate cancer, respectively [35, 36].”